# Effect of *AGTR1* A1166C genetic polymorphism on coronary artery lesions and mortality in patients with acute myocardial infarction

Duy Cong Tran[ORCID][1,2,3], Linh Hoang Gia Le[4], Truc Thanh Thai[5], Sy Van Hoang[1,2], Minh Duc Do[ORCID][4]*, Binh Quang Truong[1,3]*

**1** Department of Internal Medicine, University of Medicine and Pharmacy at Ho Chi Minh City, Ho Chi Minh City, Vietnam, **2** Department of Cardiology, Cho Ray Hospital, Ho Chi Minh City, Vietnam, **3** Cardiovascular Center, University Medical Center Ho Chi Minh City, Ho Chi Minh City, Vietnam, **4** Center for Molecular Biomedicine, University of Medicine and Pharmacy at Ho Chi Minh City, Ho Chi Minh City, Vietnam, **5** Faculty of Public Health, University of Medicine and Pharmacy at Ho Chi Minh City, Ho Chi Minh City, Vietnam

* ducminh@ump.edu.vn (MDD); binh.tq@umc.edu.vn (BQT)

**Data Availability Statement:** All relevant data are within the manuscript and its Supporting information files.

## Abstract

The pathogenesis and prognosis of patients with acute myocardial infarction (AMI) may be influenced by both genetic and environmental factors. Findings on the relationship of polymorphisms in various genes encoding the renin-angiotensin-aldosterone system with coronary artery lesions and mortality in AMI patients are inconsistent. The aim of this study was to determine whether the *AGTR1* A1166C genetic polymorphism affects coronary artery lesions and 1-year mortality in post-AMI patients. Patients with their first AMI admitted to Cho Ray Hospital, Vietnam, from January 2020 to August 2021 were enrolled in this prospective clinical study. All participants underwent invasive coronary angiography and were identified as having the genotypes of *AGTR1* A1166C by way of a polymerase chain reaction method. All patients were followed up for all-cause mortality 12 months after AMI. The association of the *AGTR1* A1166C polymorphism with coronary artery lesions and 1-year mortality was evaluated using logistic regression and Cox regression analysis, respectively. Five hundred and thirty-one AMI patients were recruited. The mean age was 63.9 ± 11.6 years, and 71.6% of the patients were male. There were no significant differences in the location and number of diseased coronary artery branches between the AA and AC+CC genotypes. The AC and CC genotypes were independently associated with $\geq$ 90% diameter stenosis of the left anterior descending (LAD) artery (odds ratio = 1.940; 95% confidence interval (CI): 1.059–3.552, $p$ = 0.032). The 1-year all-cause mortality rate difference between patients with the AC and CC genotypes versus those with the AA genotype was not statistically significant (hazard ratio = 1.000, 95% CI: 0.429–2.328, $p$ = 1.000). The *AGTR1* A1166C genetic polymorphism is associated with very severe luminal stenosis of the LAD but not with mortality in AMI patients.

**Funding:** This study was supported partially by a grant (03/2020/H-HYD) from the University of Medicine and Pharmacy at Ho Chi Minh City, Vietnam (https://ump.edu.vn/). Duy Cong Tran was funded by the Master, PhD Scholarship Program of Vingroup Innovation Foundation (VINIF) (https://vinif.org/), code VINIF.2022.TS027. The funders had no role in the study design, data collection and analysis, decision to publish, or preparation of the manuscript.

**Competing interests:** The authors have declared that no competing interests exist.

**Abbreviations:** ACEI, angiotensin-converting enzyme inhibitor; AMI, acute myocardial infarction; ARB, angiotensin II receptor blocker; AT1R, angiotensin II type 1 receptor; CAD, coronary artery disease; CI, confidence interval; DNA, Deoxyribonucleic acid; eGFR, estimated glomerular filtration rate; HR, hazard ratio; IQR, interquartile range; LAD, left anterior descending artery; LCx, left circumflex artery; LM, left main; LVEF, left ventricular ejection fraction; OR, odds ratio; PCR, polymerase chain reaction; RAAS, renin-angiotensin-aldosterone system; RCA, right coronary artery; SD, standard deviation; STEMI, ST-segment elevation myocardial infarction.

## Introduction

Coronary artery disease (CAD) is the predominant cause of death around the globe [1]. Acute myocardial infarction (AMI) is the most severe clinical manifestation of CAD. The essential mechanism of AMI is the rupture or erosion of atherosclerotic plaques, leading to thrombus formation and causing complete or incomplete occlusion of epicardial coronary artery branches [2]. In clinical practice, AMI is a common cardiovascular emergency and requires aggressive management. Although the survival status of patients with AMI has improved in recent decades, the burden of mortality and morbidity remains significant. The 1-year death rate in patients after AMI ranges from 10 to 12% [3].

Coronary artery lesions and mortality in AMI patients can be caused by both environmental and genetic factors. Many genetic markers associated with AMI have been identified, including genes encoding components of the renin-angiotensin-aldosterone system (RAAS). The *AGTR1* gene encoding the angiotensin II type 1 receptor (AT1R) is located on the long arm of chromosome 3 (3q21-25), and A1166C is the most studied polymorphism [4]. This gene variant is at the untranslated 3' region and has a base substitution of adenine with cytosine at nucleotide position 1166 of the mRNA sequence. In patients carrying the *AGTR1* A1166C genetic polymorphism, there is an increased expression of the AT1 receptor [5]. Angiotensin II, the main effector in RAAS, acts on the AT1 receptor and causes vasoconstriction, aldosterone secretion, cell proliferation, vascular remodeling, endothelial dysfunction, and atherosclerosis [6, 7].

The *AGTR1* A1166C genetic variant has been shown in previous studies to be associated with the risk of AMI. A meta-analysis of 18 case-control studies demonstrated that the C allele is a risk factor for AMI [8]. However, few studies have investigated the association between this genetic polymorphism and coronary artery lesions and mortality in patients with AMI. Kruzliak P *et al.* found that the CC genotype is associated with a higher risk of three-vessel stenosis and also a higher proportion of left anterior descending (LAD) artery infarction [9]. In contrast, when comparing the CC genotype with AC and AA genotypes in AMI patients, Araújo MA *et al.* did not detect a relationship between the *AGTR1* A1166C variant and coronary artery injuries [10]. However, *AGTR1* A1166C has been revealed in several studies to be a prognostic factor for mortality in patients with AMI [9, 11, 12], although other studies have not found any predictive value [13, 14]. Given the lack of data and the conflicting results between various studies, we conducted this study to investigate the effect of the *AGTR1* A1166C genetic variant on coronary artery lesion characteristics and mortality in Vietnamese patients with AMI.

## Materials and methods

### Study population

In this prospective clinical study, we enrolled patients who were admitted to the Department of Cardiology and the Department of Interventional Cardiology at Cho Ray Hospital in Ho Chi Minh City, Vietnam, between January 2020 and August 2021. The inclusion criteria encompassed patients who were diagnosed with their first AMI according to the fourth universal definition [2] and who agreed to participate in the study. Exclusion criteria were age below 18 years, history of AMI, percutaneous coronary intervention, and coronary artery bypass graft surgery, normal coronary angiogram, and loss of contact during the follow-up period.

The study was conducted in accordance with the principles of the Declaration of Helsinki and approved by the Ethics Committee of the University of Medicine and Pharmacy at Ho Chi Minh City (Protocol No. 550/UMP-BOARD). All participants provided written informed consent.

## Study protocol

We collected data on participants' clinical characteristics, including age, sex, clinical type of AMI, and Killip class. Traditional cardiovascular risk factors were also collected, namely hypertension (according to ESC/ESH guidelines) [15], diabetes mellitus (according to ADA guidelines) [16], dyslipidemia (according to NCEP ATP-III guidelines) [17], obesity (according to the WHO classification for the Asian population) [18], smoking, and a family history of premature CAD. Laboratory parameters were also recorded, such as Troponin I concentration, estimated glomerular filtration rate (eGFR), and left ventricular ejection fraction (LVEF). The LVEF of patients was assessed using Simpson's method for echocardiography. All patients underwent invasive coronary angiography. The characteristics of coronary artery lesions were noted, i.e., the position of diseased coronary artery branches with $\geq$ 50% stenosis of the diameter, the number of diseased coronary artery branches, and very severe stenosis of the artery coronary artery branches ($\geq$ 70% stenosis of the left main and $\geq$ 90% stenosis of the other epicardial coronary arteries). Two milliliters of venous blood were drawn from each patient for genotyping of the *AGTR1* A1166C polymorphism.

The patients were followed up during their hospital stay and up to 12 months from the date of admission. The primary outcome measured was all-cause mortality. Post-discharge follow-up was conducted through revisits and hospitalization at Cho Ray Hospital or telephone interviews.

## Genotyping of *AGTR1* A1166C

Genomic deoxyribonucleic acid (DNA) for all participants was extracted from peripheral leukocytes using the GeneJet™ Whole Blood Genomic DNA Purification Mini Kit (Thermo Fisher Scientific, Waltham, MA, USA) following the manufacturer's protocol. *AGTR1* A1166C was genotyped for each sample by two separate polymerase chain reactions (PCR). Each reaction used three primers with similar melting temperatures to determine the presence of allele A or C based on the appearance of a 342-basepair DNA band on electrophoresis. All the PCRs were performed using a SimpliAmp™ Thermal Cycler (Thermo Fisher Scientific). The details of the PCR primers, components, and conditions are described in the Supplementary materials.

Twenty DNA samples were randomly chosen and directly sequenced using appropriate primers and conditions in an ABI 3500 sequencer (Thermo Fisher Scientific). The protocol for direct sequencing has been described previously [19–22]. The DNA samples with an identified genotype were used as positive controls for PCR.

## Statistical analysis

Qualitative variables were presented as frequencies and percentages. The Kolmogorov-Smirnov test was used to check the normal distribution of quantitative variables, expressed as mean with standard deviation (SD) if normally distributed or median with interquartile range [IQR] if not normally distributed. Consistency with the Hardy-Weinberg equilibrium of genotypes was assessed using Chi-squared tests. Differences in the characteristics of coronary artery lesions between *AGTR1* A1166C genetic polymorphism genotypes were evaluated using Chi-squared or Fisher's exact tests. Factors affecting the degree of $\geq$ 90% stenosis of the LAD diameter were assessed by odds ratio (OR) and 95% CI in univariate and multivariable logistic regression analyses. Differences in the genotypes of the *AGTR1* A1166C polymorphism and other factors between those who survived and those who did not were compared using Chi-squared or Fisher's exact tests for qualitative variables and Student's t-test or the Mann-Whitney U test based on the data distribution. Factors associated with 1-year all-cause mortality in the univariate analysis were analyzed in a multivariable Cox regression model to identify independent prognostic factors.

Statistical analysis was performed using SPSS Statistics for Windows version 22.0 (IBM Corp., Armonk, NY, USA). A *p*-value of less than 0.05 was considered to be statistically significant.

## Results

### Characteristics of the study population

This study included 531 patients with their first AMI, most of whom were male (71.6%) (Table 1). The mean age of the participants was 63.9 ± 11.6 years. ST-segment elevation AMI

**Table 1. Baseline characteristics of the study population.**

| Variables | Characteristic (n = 531) |
|---|---|
| **Clinical and laboratory parameters** | |
| Age (years) | 63.9 ± 11.6 |
| Male | 380 (71.6%) |
| Female | 151 (28.4%) |
| STEMI | 336 (63.3%) |
| Killip Class | |
| Class I | 406 (76.5%) |
| Class II | 44 (8.3%) |
| Class III | 32 (6.0%) |
| Class IV | 49 (9.2%) |
| Admission Troponin I (pg/mL) | 15.1 (2.9–50.0) |
| eGFR (mL/min/1.73 m$^2$) | 83.2 (64.3–94.5) |
| LVEF (%) | 46.0 (39.0–53.0) |
| **Coronary artery lesions** | |
| Diseased LM | 53 (10.0%) |
| Diseased LAD | 473 (89.1%) |
| Diseased LCx | 296 (55.7%) |
| Diseased RCA | 380 (71.6%) |
| One-vessel disease | 129 (24.3%) |
| Two-vessel disease | 186 (35.0%) |
| Three-vessel disease | 216 (40.7%) |
| ≥ 70% LM stenosis | 25 (4.7%) |
| ≥ 90% LAD stenosis | 254 (47.8%) |
| ≥ 90% LCx stenosis | 123 (23.2%) |
| ≥ 90% RCA stenosis | 234 (44.1%) |
| **Treatment** | |
| Coronary revascularization | 503 (94.7%) |
| Aspirin | 530 (99.8%) |
| P2Y12 inhibitor | 531 (100.0%) |
| Statin | 523 (98.5%) |
| ACEI/ARB | 480 (90.4%) |
| Beta-blocker | 406 (76.5%) |

Values are presented as the mean ± SD, number (%) or median (interquartile range). STEMI, ST-segment elevation myocardial infarction; eGFR, estimated glomerular filtration rate; LVEF, left ventricular ejection fraction; LM, left main; LAD, left anterior descending artery; LCx, left circumflex artery; RCA, right coronary artery; ACEI, angiotensin-converting enzyme inhibitor; ARB, angiotensin II receptor blocker.

**Table 2. Genotypes of *AGTR1* A1166C polymorphism and traditional cardiovascular risk factors.**

| Variables | Frequency (%) (n = 531) |
|---|---|
| **Genotypes of *AGTR1* A1166C polymorphism** | |
| AA | 476 (89.7%) |
| AC | 49 (9.2%) |
| CC | 6 (1.1%) |
| **Traditional cardiovascular risk factors** | |
| Dyslipidemia | 476 (89.6%) |
| Hypertension | 435 (81.9%) |
| Smoking | 224 (42.2%) |
| Diabetes mellitus | 130 (24.5%) |
| Obesity | 108 (20.3%) |
| Family history of premature CAD | 35 (6.6%) |

CAD, coronary artery disease.

was diagnosed in 63.3% of patients. Regarding the degree of acute heart failure, the proportion of patients with Killip class $\geq$ II was 23.5%. The prevalent diseased coronary artery was the LAD (89.1%). Three-vessel disease was predominant (40.7%) in coronary angiography. The majority of patients underwent coronary revascularization (94.7%) with fibrinolysis, percutaneous coronary intervention, or coronary bypass surgery. In addition, participants received standard medical therapy with aspirin (99.8%), P2Y12 inhibitor (100.0%), statin (98.5%), ACEI/ARB (90.4%), and beta-blocker (76.5%).

The genotypes of the *AGTR1* A1166C polymorphism in our study were not in agreement with the Hardy-Weinberg equilibrium ($p = 0.01$) (Table 2). The AA genotype was most common in AMI patients (89.7%). In terms of traditional cardiovascular risk factors, dyslipidemia was the most frequent factor (89.6%), and a family history of premature CAD was not common (6.6%).

## Association of *AGTR1* A1166C genetic polymorphism with coronary artery lesions in AMI patients

There was no association between the *AGTR1* A1166C genetic variant and the location of diseased coronary artery branches, i.e., LM, LAD, LCx, and RCA (Table 3). No significant differences in the number of diseased coronary artery branches were found between patients carrying the AA genotype and those with the AC and CC genotypes. However, the univariate analysis showed that patients with the AC and CC genotypes had a greater risk of $\geq$ 90% LAD stenosis than those with the AA genotype ($p = 0.028$). Univariate logistic regression analysis also showed that obesity and left ventricular ejection fraction were related to very severe stenosis of the LAD ($p = 0.022$ and $p < 0.001$, respectively). After adjusting for confounding factors in the multivariable logistic regression analysis, the AC and CC genotype was found to be independently associated with very severe stenosis of the LAD in AMI patients (OR = 1.940; 95% CI: 1.059–3.552; $p = 0.032$) (Table 4).

## Association of the *AGTR1* A1166C genetic polymorphism and other factors with 1-year all-cause mortality in AMI patients

There were 58 all-cause deaths during the 1-year follow-up (10.9%), including 52 patients with the AA genotype and 6 patients with the AC or CC genotype (Table 5). In univariate analysis,

**Table 3. Association between *AGTR1* A1166C genetic polymorphism and coronary artery lesions.**

| Variables | AA (n = 476) | AC+CC (n = 55) | *p*-value |
|---|---|---|---|
| Diseased LM | 46 (9.7%) | 7 (12.7%) | 0.473 |
| Diseased LAD | 422 (88.7%) | 51 (92.7%) | 0.359 |
| Diseased LCx | 264 (55.5%) | 32 (58.2%) | 0.701 |
| Diseased RCA | 342 (71.8%) | 38 (69.1%) | 0.668 |
| One-vessel disease | 116 (24.4%) | 13 (23.6%) | 0.904 |
| Two-vessel disease | 168 (35.3%) | 18 (32.7%) | 0.706 |
| Three-vessel disease | 192 (40.3%) | 24 (43.6%) | 0.637 |
| $\geq$ 70% LM stenosis | 21 (4.4%) | 4 (7.3%) | 0.314 |
| $\geq$ 90% LAD stenosis | 220 (46.2%) | 34 (61.8%) | 0.028 |
| $\geq$ 90% LCx stenosis | 110 (23.1%) | 13 (23.6%) | 0.930 |
| $\geq$ 90% RCA stenosis | 215 (45.2%) | 19 (34.5%) | 0.133 |

LM, left main; LAD, left anterior descending artery; LCx, left circumflex artery; RCA, right coronary artery.

diabetes mellitus, age, Killip class $\geq$ II, Troponin I on admission, eGFR, LVEF, and ACEI/ARB therapy were found to be associated with 1-year all-cause mortality. The *AGTR1* A1166C genetic polymorphism had no effect on 1-year all-cause mortality, with HR = 1.000 (95% CI: 0.429–2.328; *p* = 1.000) in comparison between the AC+CC and AA genotypes. After adjusting for other factors in the multivariable Cox regression analysis, independent prognostic factors for 1-year all-cause mortality were Killip class $\geq$ II (HR = 2.375, 95% CI: 1.352–4.171, *p* = 0.003), admission Troponin I (HR = 1.001, 95% CI: 1.000–1.003, *p* = 0.047), and eGFR (HR = 0.985, 95% CI: 0.973–0.998, *p* = 0.020) (Table 6).

## Discussion

The genotypes of the *AGTR1* A1166C polymorphism in our study were not in agreement with the Hardy-Weinberg equilibrium, possibly because our study population was a group of

**Table 4. Effect of the *AGTR1* A1166C genetic polymorphism and other factors on very severe stenosis of the left anterior descending artery.**

| Variables | Unadjusted OR (95% CI) | *p*-value | Adjusted OR (95% CI) | *p*-value |
|---|---|---|---|---|
| AC+CC vs. AA | 1.884 (1.062–3.342) | 0.030 | 1.940 (1.059–3.552) | 0.032 |
| Dyslipidemia | 1.025 (0.586–1.794) | 0.930 | | |
| Hypertension | 1.048 (0.673–1.632) | 0.835 | | |
| Smoking | 0.853 (0.604–1.204) | 0.365 | | |
| Diabetes mellitus | 0.843 (0.566–1.254) | 0.398 | | |
| Obesity | 0.603 (0.391–0.930) | 0.022 | 0.647 (0.409–1.021) | 0.062 |
| Family history of premature CAD | 1.167 (0.588–2.316) | 0.660 | | |
| Age (years) | 1.014 (0.999–1.029) | 0.073 | | |
| Male | 0.870 (0.596–1.268) | 0.468 | | |
| STEMI | 1.353 (0.949–1.930) | 0.095 | | |
| Killip Class $\geq$ II | 1.075 (0.720–1.603) | 0.724 | | |
| Admission Troponin I (pg/mL) | 1.000 (0.997–1.002) | 0.798 | | |
| eGFR (mL/min/1.73 m$^2$) | 0.999 (0.992–1.006) | 0.798 | | |
| LVEF | 0.940 (0.923–0.957) | <0.001 | 0.940 (0.924–0.957) | <0.001 |

OR, odds ratio; CI, confidence interval; CAD, coronary artery disease; STEMI, ST-segment elevation myocardial infarction; eGFR, estimated glomerular filtration rate; LVEF, left ventricular ejection fraction.

**Table 5. Association of the *AGTR1* A1166C genetic polymorphism and other factors with 1-year all-cause mortality in AMI patients.**

| Variables | Death (n = 58) | Survivor (n = 473) | *p*-value |
|---|---|---|---|
| **Genotypes of *AGTR1* A1166C polymorphism** | | | |
| AA | 52 (89.7%) | 424 (89.6%) | 0.997 |
| AC+CC | 6 (10.3%) | 49 (10.4%) | |
| **Traditional cardiovascular risk factors** | | | |
| Dyslipidemia | 52 (89.7%) | 424 (89.6%) | 0.997 |
| Hypertension | 48 (82.8%) | 387 (81.8%) | 0.861 |
| Smoking | 22 (37.9%) | 202 (42.7%) | 0.487 |
| Diabetes mellitus | 22 (37.9%) | 108 (22.8%) | 0.012 |
| Obesity | 10 (17.2%) | 98 (20.7%) | 0.535 |
| Family history of premature CAD | 4 (6.9%) | 31 (6.6%) | 0.785 |
| **Clinical and laboratory factors** | | | |
| Age (years) | 67.5±10.5 | 63.5±11.7 | 0.013 |
| Male | 36 (62.1%) | 344 (72.7%) | 0.089 |
| STEMI | 38 (65.5%) | 298 (63.0%) | 0.708 |
| Killip Class $\geq$ II | 29 (50.0%) | 97 (20.5%) | <0.001 |
| Admission Troponin I (pg/mL) | 37.5 (3.26–50.0) | 13.9 (2.7–50.0) | 0.029 |
| eGFR (mL/min/1.73 m$^2$) | 62.1 (40.5–86.7) | 84.6 (67.7–95.0) | <0.001 |
| LVEF (%) | 40.5 (31.5–50.0) | 47.0 (40.0–53.0) | 0.002 |
| **Coronary artery lesions** | | | |
| Diseased LM | 7 (12.1%) | 46 (9.7%) | 0.574 |
| Diseased LAD | 53 (91.4%) | 420 (88.8%) | 0.551 |
| Diseased LCx | 32 (55.2%) | 264 (55.8%) | 0.926 |
| Diseased RCA | 42 (72.4%) | 338 (71.5%) | 0.879 |
| One-vessel disease | 12 (20.7%) | 117 (24.7%) | 0.498 |
| Two-vessel disease | 23 (39.7%) | 163 (34.5%) | 0.434 |
| Three-vessel disease | 23 (29.7%) | 193 (40.8%) | 0.867 |
| $\geq$ 70% LM stenosis | 5 (8.6%) | 20 (4.2%) | 0.177 |
| $\geq$ 90% LAD stenosis | 30 (51.7%) | 224 (47.4%) | 0.530 |
| $\geq$ 90% LCx stenosis | 18 (31.0%) | 105 (22.2%) | 0.132 |
| $\geq$ 90% RCA stenosis | 27 (46.6%) | 207 (43.8%) | 0.686 |
| **Treatment** | | | |
| Coronary revascularization | 53 (91.4%) | 450 (95.1%) | 0.216 |
| Aspirin | 57 (98.3%) | 473 (100.0%) | 0.109 |
| P2Y12 inhibitor | 58 (100.0%) | 473 (100.0%) | - |
| Statin | 57 (98.3%) | 466 (98.5%) | 0.606 |
| ACEI/ARB | 47 (81.0%) | 433 (91.5%) | 0.010 |
| Beta-blocker | 43 (74.1%) | 363 (76.7%) | 0.659 |

Values are presented as number (%) or median (interquartile range). CAD, coronary artery disease; STEMI, ST-segment elevation myocardial infarction; eGFR, estimated glomerular filtration rate; LVEF, left ventricular ejection fraction; LM, left main; LAD, left anterior descending artery; LCx, left circumflex artery; RCA, right coronary artery; ACEI, angiotensin-converting enzyme inhibitor; ARB, angiotensin II receptor blocker.

patients with AMI, not the entire population of case and control groups as in case-control studies. We did not select a control group because doing so was not necessary to answer the research question of whether the *AGTR1* A1166C variant is associated with coronary artery lesions and mortality in AMI patients. The CC genotype was the least common (1.1%) in our study population. This feature is consistent with other studies on different races, not just on Asian populations [9, 10, 23–25].

**Table 6. Factors associated with 1-year all-cause mortality in multivariable Cox regression analysis.**

| Variables | HR | 95% CI | *p*-value |
|---|---|---|---|
| Diabetes mellitus | 1.338 | 0.761–2.352 | 0.312 |
| Age (years) | 1.013 | 0.988–1.038 | 0.309 |
| Killip Class $\geq$ II | 2.375 | 1.352–4.171 | 0.003 |
| Admission Troponin I (pg/mL) | 1.001 | 1.000–1.003 | 0.047 |
| eGFR (mL/min/1.73 m$^2$) | 0.985 | 0.973–0.998 | 0.020 |
| LVEF (%) | 0.977 | 0.954–1.002 | 0.068 |
| ACEI/ARB | 0.662 | 0.326–1.343 | 0.253 |

HR, hazard ratio; CI, confidence interval; eGFR, estimated glomerular filtration rate; LVEF, left ventricular ejection fraction; ACEI, angiotensin-converting enzyme inhibitor; ARB, angiotensin II receptor blocker.

Our study showed that patients with the AC and CC genotypes had a higher rate of $\geq$ LAD 90% stenosis than AA genotype carriers. The study by Kruzliak P *et al.* demonstrated that acute coronary syndrome patients carrying the CC genotype in the Slovak Republic had a 4.08-fold higher risk of LAD infarction and a 3.87 times higher risk of three-vessel disease compared with those having the AA and AC genotypes [9]. Nevertheless, Araújo MA *et al.* found no differences in the number of diseased coronary vessels and morphological features of the atherosclerotic plaque among the *AGTR1* A1166C genotypes [10]. The results of other studies are conflicting regarding the effect of *AGTR1* A1166C variants on coronary artery lesions due to differences in genotype distributions across countries and races, and the frequency of environmental factors affecting coronary atherosclerosis. In terms of pathogenesis, the AT1 receptor may be overexpressed in carriers of the *AGTR1* A1166C variant, leading to increased adverse effects of angiotensin II on coronary atherosclerosis. Angiotensin II promotes coronary artery injury possibly through various pathophysiological phenomena, such as modulation of the inflammatory response, stimulation of the production of multiple cytokines, such as IL-6, TNF-α, and COX-2, and promotion of the generation of reactive oxygen species, enhanced oxidative stress, decreased nitric oxide production, and increased endothelial dysfunction [3].

The *AGTR1* A1166C genetic polymorphism was not found to be associated with 1-year all-cause mortality in AMI patients in the present study. Similar results have been observed in other studies [13, 14]. The CC genotype was not found to be associated with in-hospital mortality in the GEMIG (Genetics and Epidemiology of Acute Myocardial Infarction in the Greek Population) study in Greece [14]. A study by Brsic E *et al.* involving young Italian patients with AMI concluded that the *AGTR1* A1166C polymorphism was not related to the composite end points of major cardiovascular events, including cardiovascular mortality, myocardial infarction, and revascularization procedures during the follow-up period of 46 ± 12 months [13].

In contrast, the *AGTR1* A1166C genetic polymorphism has been shown in several studies to be a predictor of mortality in AMI patients [9, 11, 12]. Kruzliak P *et al.* found that the CC genotype is associated with a 6.48-fold increased risk of sudden cardiac death within 24 hours of emergency room admission in patients with acute coronary syndromes [9]. After adjusting for the cause of death, the CC genotype was found in a large prospective study in France to be an independent predictor of post-AMI cardiovascular mortality during follow-up (median 2.5 years) [11]. In addition, a study by Franco E *et al.* concluded that the AC genotype was associated with death, new myocardial infarction, and coronary revascularization in young Italian patients with AMI during the follow-up period of 9 ± 4 years [12].

The inconsistency in the results of various studies on the influence of the *AGTR1* A1166C genetic variant on mortality in AMI patients could be due to differences in the occurrence of the *AGRT1* A1166C genotypes across distinct ethnicities, countries, and geographic regions, as well as differences in study populations, study designs, and survival follow-up periods. In addition, ACEI and ARB have been shown in previous studies to improve mortality in patients after AMI [26–28]. Therefore, the impact of ACEI and ARB might obscure the predictive value of the *AGTR1* A1166C polymorphism for mortality. The proportion of patients using ACEI/ARB in our study was very high (90.4%).

Our study had some limitations that should be taken into consideration. First, this study was conducted in only one center, so it might not represent the genotype characteristics of the *AGTR1* A1166C variant in Vietnam. Next, the pathophysiology of AMI includes the interaction of many factors and genes, so the *AGTR1* A1166C might not fully explain the association of this polymorphism with coronary artery lesions and mortality in patients with AMI. Finally, the study was observational and thus did not intervene in treatment strategies and adherence to therapy in AMI patients.

## Conclusions

The *AGTR1* A1166C genetic polymorphism is associated with very severe luminal stenosis of the LAD. Nevertheless, it is not associated with 1-year all-cause mortality in Vietnamese patients after AMI. These results contribute to the genetic assessment of coronary artery lesions. However, further studies with larger sample sizes and more extended follow-up periods are needed to consider the prognostic role of the *AGTR1* A1166C variant in patients with AMI.

## Supporting information

**S1 File. PCR primers, components, and conditions for AGTR1 A1166C genotyping.** (DOCX)

**S1 Checklist. Human participants research checklist.** (DOCX)

**S1 Data.** (XLSX)

## Author Contributions

**Conceptualization:** Duy Cong Tran, Truc Thanh Thai, Sy Van Hoang, Minh Duc Do, Binh Quang Truong.

**Data curation:** Duy Cong Tran, Linh Hoang Gia Le, Minh Duc Do.

**Formal analysis:** Duy Cong Tran, Truc Thanh Thai, Minh Duc Do.

**Funding acquisition:** Duy Cong Tran.

**Investigation:** Duy Cong Tran, Linh Hoang Gia Le, Sy Van Hoang, Minh Duc Do, Binh Quang Truong.

**Methodology:** Truc Thanh Thai, Binh Quang Truong.

**Writing – original draft:** Duy Cong Tran.

**Writing – review & editing:** Duy Cong Tran, Truc Thanh Thai, Minh Duc Do, Binh Quang Truong.

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
