## [Decision Letter · Decision Letter 0]

5 Feb 2024

PONE-D-23-38605Effect of AGTR1 A1166C Genetic Polymorphism on Coronary Artery Lesions and Mortality in Patients with Acute Myocardial InfarctionPLOS ONE

Dear Dr. Do,

Thank you for submitting your manuscript to PLOS ONE. After careful consideration, we feel that it has merit but does not fully meet PLOS ONE’s publication criteria as it currently stands. Therefore, we invite you to submit a revised version of the manuscript that addresses the points raised during the review process.

We look forward to receiving your revised manuscript.

Kind regards,

Eyüp Serhat Çalık

Academic Editor

PLOS ONE

Journal Requirements:

"University of Medicine and Pharmacy at Ho Chi Minh City."

4. Thank you for stating the following in your Competing Interests section: No

**Additional Editor Comments:**

Dear Authors

I read with interest your manuscript exploring this current and interesting topic. It was reviewed by three peer reviewers. Their recommendations are below and in the attached copy. Please respond to the reviewers' questions point by point and resubmit your manuscript after major revision, especially by adding a control group of normal individuals as suggested by reviewer 3. I wish you every success.

Reviewers' comments:

Reviewer's Responses to Questions

**Comments to the Author**

1. Is the manuscript technically sound, and do the data support the conclusions?

Reviewer #1: Partly

Reviewer #2: No

Reviewer #3: No

2. Has the statistical analysis been performed appropriately and rigorously? 

Reviewer #1: Yes

Reviewer #2: No

Reviewer #3: No

3. Have the authors made all data underlying the findings in their manuscript fully available?

Reviewer #1: Yes

Reviewer #2: Yes

Reviewer #3: Yes

4. Is the manuscript presented in an intelligible fashion and written in standard English?

Reviewer #1: Yes

Reviewer #2: No

Reviewer #3: Yes

5. Review Comments to the Author

Reviewer #1: Dear Author(s),

After fully and carfully reading the manuscript, I have some points as bellow.

1. I have minor correction fixed on the manuscript.

2. I have a question related to the table 4.

good luck

Reviewer #2: The manuscript addresses an important clinical question related to the genetics of AMI and its associated outcomes. However, the manuscript contains fundamental errors that cannot be rectified through author revisions. Specifically, the number of specific subjects is insufficient for this study.

Reviewer #3: First comment: The study design seems bizarre and not sound.

The design and analysis of genetic polymorphisms studies generally involve relating a particular disease or health outcome to a particular exposure or genetic trait, while assessing the presence of systematic error, controlling random error and assessing effect modification (interaction) with other exposures or traits.

To achieve these requirements, there should be two distinct groups to compare -usually -normal controls are used in this type of studies as comparative group.

Also, the authors stated in the discussion that “they did not select a control group because doing so was not necessary to answer the research question of whether the AGTR1 A1166C variant is associated with coronary artery lesions and mortality in AMI patients”. How they proved that this variant is associated with the lesions and mortality without comparing it with normal population?

What if the variant prevalence in normal population is similar to their current results?

Second comment: How the author calculated the Hardy-Weinberg equilibrium without having two groups (cases and controls)?

Third comment: On what basis they used genotypes AC+CC vs. AA in most of their comparisons?

Fourth comment: There are many grammatical and prepositions errors throughout the manuscript, and it needs major English Language revision.

6. PLOS authors have the option to publish the peer review history of their article (what does this mean?). If published, this will include your full peer review and any attached files.

Reviewer #1: **Yes: **Dhafer A. F. Al-Koofee

Reviewer #2: No

Reviewer #3: **Yes: **Hoda Y. Abdallah

---

## [Author Response · Author response to Decision Letter 0]

13 Feb 2024

RESPONSE TO REVIEWERS

Reviewer #1:

1. I have minor correction fixed on the manuscript.

Thank you for your careful reading and correction. We corrected several points according to your suggestion on the manuscript.

2. I have a question related to the table 4.

We calculated the odds ratio of the association between the AGTR1 A1166C genetic polymorphism and very severe stenosis of the left anterior descending artery (LAD) among acute myocardial infarction patients based on the number of participants in groups listed in the following table.

Group Very severe stenosis of LAD No severe stenosis of LAD Total

AC+CC 34 21 55

AA 220 256 476

Total 254 277 531

Reviewer #2: The manuscript addresses an important clinical question related to the genetics of AMI and its associated outcomes. However, the manuscript contains fundamental errors that cannot be rectified through author revisions. Specifically, the number of specific subjects is insufficient for this study.

Thank you for your comment. The initial sample size was calculated to estimate the difference in the survival rates among different genotypes of AGTR1 A1166C polymorphism. A minimum of 33 deaths or 413 AMI patients were needed to have a statistical power of at least 80% with a 95% confidence interval to detect a hazard ratio of 2 or more. In reality, there were 531 AMI patients enrolled and 58 all-cause deaths during the 1-year follow-up in our study.

Reviewer #3: 

We would like to thank you Reviewer 3 for the detailed comments for our manuscript. We would like to address all the comments point by point:

First comment: The study design seems bizarre and not sound.

The design and analysis of genetic polymorphisms studies generally involve relating a particular disease or health outcome to a particular exposure or genetic trait, while assessing the presence of systematic error, controlling random error and assessing effect modification (interaction) with other exposures or traits.

To achieve these requirements, there should be two distinct groups to compare -usually -normal controls are used in this type of studies as comparative group.

Also, the authors stated in the discussion that “they did not select a control group because doing so was not necessary to answer the research question of whether the AGTR1 A1166C variant is associated with coronary artery lesions and mortality in AMI patients”. How they proved that this variant is associated with the lesions and mortality without comparing it with normal population?

What if the variant prevalence in normal population is similar to their current results?

Despite the diverse evidence of genetic variants in the renin-angiotensin-aldosterone system in primary prevention setttings of cardiovascular diseases such as acute myocardial infarction, whether it affects coronary artery lesions and mortality in patients diagnosed with AMI in the secondary prevention setting has not been well studied and remains controversial. A case-control study design is needed to determine the association between AGTR1 A1166C genetic polymorphism and the risk of AMI in primary prevention settings. Nevertheless, we aimed to investigate the association of AGTR1 A1166C genotypes with coronary artery lesions and mortality in patients with AMI in the secondary prevention setting. Therefore, the control group might not be needed to answer the research question in our study. Similarly, several previous studies by Tokunaga S et al (2001) (ref: doi: 10.2143/AC.56.6.2005698), Palmer BR et al (2003) (ref: doi: 10.1016/s0735-1097(02)02927-3), Goldenberg I et al (2006) (ref: doi: 10.1161/01.HYP.0000239204.41079.6b), Hara M et al, OACIS Investigators (2014) (ref: doi: 10.1536/ihj.13-288), Martínez-Quintana E et al (2014) (ref: doi: 10.5603/CJ.a2013.0107) and Moorthy N et al (2021) (ref: doi: 10.1016/j.ijcha.2020.100701) also did not recruit a control group. Therefore, we did not select a control group, but compared groups with different genotypes of AGTR1 A1166C polymorphism among AMI patients.

Second comment: How the author calculated the Hardy-Weinberg equilibrium without having two groups (cases and controls)?

The Hardy-Weinberg equilibrium can be calculated in either cases or controls. Thus, the Hardy-Weinberg equilibrium of genotypes of the AGTR1 A1166C polymorphism in AMI patients was calculated using Chi-squared test in SPSS Statistics for Windows version 22.0 (IBM Corp., Armonk, NY, USA) based on observed and expected numbers of participants in different genotype groups. 

Genotype group Observed number of participiants Expected number of participiants

AA 476 472

AC 49 57

CC 6 2

AGTR1 A1166C genotypes in our study population were not at Hardy-Weinberg equilibrium (P < 0.05). This could be explained by the fact that our study included only AMI patients. Deviation from Hardy-Weinberg equilibrium can also be caused by selection bias or error in genotyping. Nevertheless, selection bias was tightly controlled by accurately selecting AMI subjects who satisfied the sampling criteria through careful history taking, thorough clinical examination, and performing all necessary tests to diagnose AMI definitely. In addition, the genetic testing in this study was performed according to precise and rigorous procedures.

Third comment: On what basis they used genotypes AC+CC vs. AA in most of their comparisons?

 As the number of paticipants with AGTR1 CC genotype was rare and alen C was consider “the risk group” according to previous studies in the literature, we used the recessive genetic model to compare AC+CC vsersus AA in terms of coronary artery lesions and mortality in AMI patients.

Fourth comment: There are many grammatical and prepositions errors throughout the manuscript, and it needs major English Language revision.

We revised grammatical and prepositions errors in our manuscript.

Once again, we would like to thank Reviewer 3 for your time in reviewing our manuscript.

REFERENCE:

1. Tokunaga S, Tsuji H, Nishiue T, et al. Lower mortality in patients with the DD genotype of the angiotensin-converting enzyme gene after acute myocardial infarction. Acta Cardiol. 2001;56:351-5. doi: 10.2143/AC.56.6.2005698.

2. Palmer BR, Pilbrow AP, Yandle TG, et al. Angiotensin-converting enzyme gene polymorphism interacts with left ventricular ejection fraction and brain natriuretic peptide levels to predict mortality after myocardial infarction. J Am Coll Cardiol. 2003;41(50):729-36. doi: 10.1016/s0735-1097(02)02927-3.

3. Goldenberg I, Moss AJ, Ryan D, et al. Polymorphism in the angiotensinogen gene, hypertension, and ethnic differences in the risk of recurrent coronary events. Hypertension. 2006;48:693-699. doi: 10.1161/01.HYP.0000239204.41079.6b.

4. Hara M, Sakata Y, Nakatani D, et al. Renin-angiotensin-aldosterone system polymorphism and 5-year mortality in survivors of acute myocardial infarction. Int Heart J. 2014;55:190-196. doi: 10.1536/ihj.13-288. doi: 10.5603/CJ.a2013.0107.

5. Martinez-Quintana E, Chirino R, Nieto-Lago V, et al. Prognostic value of ACE I/D, AT1R A1166C, PAI-I 4G/5G and GPIIIa a1/a2 polymorphisms in myocardial infarction. Cardiol J. 2014;21(3):229-237.

6. Moorthy N, Ramegowda KS, Jain S, et al. Role of Angiotensin-Converting Enzyme (ACE) gene polymorphism and ACE activity in predicting outcome after acute myocardial infarction. Int J Cardiol Heart Vasc. 2021;32:100701. doi: 10.1016/j.ijcha.2020.100701.

---

## [Editor Report · Decision Letter 1]

26 Feb 2024

Effect of AGTR1 A1166C Genetic Polymorphism on Coronary Artery Lesions and Mortality in Patients with Acute Myocardial Infarction

PONE-D-23-38605R1

Dear Dr. Do,

We’re pleased to inform you that your manuscript has been judged scientifically suitable for publication and will be formally accepted for publication once it meets all outstanding technical requirements.

Kind regards,

Eyüp Serhat Çalık

Academic Editor

PLOS ONE
---

## [Editor Report · Acceptance letter]

28 Mar 2024

PONE-D-23-38605R1 

PLOS ONE

Dear Dr. Do, 

I'm pleased to inform you that your manuscript has been deemed suitable for publication in PLOS ONE. Congratulations! Your manuscript is now being handed over to our production team.

Kind regards, 

on behalf of

Dr. Eyüp Serhat Çalık 

Academic Editor

PLOS ONE